# Impact of Types of Breathing on Static Balance Ability in Healthy Adults

**DOI:** 10.3390/ijerph19031205

**Published:** 2022-01-21

**Authors:** Sung-Hyeon Kim, Ho-Jin Shin, Hwi-Young Cho

**Affiliations:** 1Department of Health Science, Gachon University Graduate School, Incheon 21936, Korea; gpgkorea30@gmail.com (S.-H.K.); gpgkorea89@gmail.com (H.-J.S.); 2Department of Physical Therapy, College of Health Science, Gachon University, Incheon 21936, Korea

**Keywords:** healthy volunteers, breathing, balance ability

## Abstract

Recent studies have suggested that breathing type may affect balance ability. However, most of these studies were conducted on the elderly and patients with musculoskeletal or neurological disorders. Therefore, the effect of voluntary breathing, such as thoracic and abdominal breathing, on the balance ability of people in various age groups is not clearly understood. The purpose of this study was to investigate the differences in balance ability according to the type of breathing in healthy young adults. This study included 78 healthy, young adults. All subjects were assessed for balance ability in neutral breathing, thoracic breathing, and abdominal breathing through a crossover design. Balance ability was assessed during static standing using a force plate. Participants were trained in voluntary breathing, evaluated using electromyography. During voluntary breathing, sway velocity, anterior-posterior difference, and anterior-posterior standard deviation increased while anterior-posterior sample entropy decreased compared to neutral breathing (*p* < 0.05). Compared with thoracic breathing, abdominal breathing increased sway velocity and variability, and reduced complexity (*p* < 0.05). These findings show that balance ability is affected by breathing, even in healthy young adults.

## 1. Introduction

Balance is the ability to maintain the center of gravity (COG) in the base of support (BOS) with minimum postural sway [1,2,3,4,5]. Balance ability is essential to perform all the movements required for activities of daily living. Sitting, standing, two-legged and one-legged standing, and gait are the most basic and functional movements in daily life based on balance function. Deterioration of body functions due to aging and diseases could adversely affect balance functions, such as a decrease in the ability to maintain COG in the BOS and an increase in the sway area in response to postural changes [3,4,5]. Furthermore, this decrease in balance ability can lead to a decrease in the range of daily life and an increase in the fall accident rate [6].

To maintain balance, the ability to perceive the body through various sensory organs is required [7]. The visual system provides sensory information to maintain balance and perception of position and direction in space [8]. The somatosensory system provides sensory information through joint receptors, muscle spindles, Golgi tendon organs, and cutaneous receptors. The vestibular system provides information about the position of the head and receives movement information related to gravity and inertial forces. Information about body position is transmitted to the central nervous system, which transmits signals to the muscular system to maintain balance. The muscular system maintains the human body’s equilibrium by controlling posture through signals from the central nervous system [9,10]. As such, the maintenance and control of balance are required for the complex coordination of the sensory system, central nervous system, and muscular system as well as coordination with other physical factors.

In order to accurately evaluate balance ability, a previous study has suggested that the environment between measurements or between subjects be controlled, and generally recommend adjusting the ground and the subject’s measurement posture [11]. In addition to these standardized evaluation methods, some studies have recently reported that breathing can also affect balance ability [12,13]. Voluntary breathing can activate the cerebral cortex and, along with it, can affect muscle control in the lower extremities and upper extremities [12]. Hernandez et al. [12] suggested that the degree of synchronization of breathing and postural sway is high in the elderly, and therefore voluntary breathing has a great influence on the maintenance and control of balance. Alain et al. [13] suggested that thoracic movements have a longer lever arm than abdominal movements, and thoracic breathing (TB) may induce greater sway than abdominal breathing (AB) because it induces activation of peripheral muscles around the cervical spine.

Balance ability is most commonly considered to evaluate the motor function, daily life movements, gait, and sports performance in various subjects [14], and understanding of balance ability is essential to accurately perform it. The maintenance and control of balance could be affected not only by visual, vestibular senses, and proprioception, but also by breathing and other factors, but there are very few studies that clearly investigate this. In addition, most of the studies that have studied balance to date have been conducted in the elderly and patients who are highly affected by variables related to breathing. For this reason, the effects of breathing methods on different age groups and healthy people are not clear. The purpose of this study was to investigate the differences in balance ability according to the type of breathing in healthy young adults.

## 2. Materials and Methods

### 2.1. Participants

Participants in this study were recruited through bulletin board advertisements and posters in community service centers and G university in I city. Participants were recruited for sedentary residents who did not a vigorous-intensity physical activity for 3 or more days per week, for 30 or more minutes [12,15]. The exclusion criteria were as follows: (1) smokers and those with abnormal lung function; (2) neurological or musculoskeletal disorders; and (3) inability to maintain a one-legged standing position for more than 20 s. A total of 94 volunteers expressed their intention to participate in the study, and 78 healthy young adults (mean age: 22.54 years) who met the study participation criteria were enrolled. All participants signed the research consent form after a detailed explanation of the study process, benefits, and risks of side effects. This study was approved by the Institutional Review Board of Gachon University (IRB number: 1044396-202101-HR-003-01) and was registered in a clinical research information service that complies with the World Health Organization International Clinical Trials Registry Platform (registration number: KCT0006026).

The sample size was calculated using G-power software (version 3.1.9.4, Heinrich Heine University, Dusseldorf, Germany) [16,17]. Based on the study results of Hernandez et al. [12], the effect size f was set to 0.246, the alpha level was set to 0.05, and the power was set to 0.8. Consequently, a sample size of 29 was required. Considering a dropout rate of 20%, a total of 37 participants were required.

### 2.2. Study Design

This study followed a randomized crossover design. A CONSORT flow diagram with a crossover design is shown in Figure 1. This study evaluated balance ability in three breathing conditions (neutral breathing [NB], TB, and AB). After baseline (NB) measurements, 78 subjects were randomized to the TB group or AB group using a permuted-block randomization method. Participants received breathing training corresponding to each group. Balance ability was assessed using the trained breathing technique. After a one-week washout period, all participants were trained in the rest of the breathing methods, and their balance ability was assessed using the trained breathing methods. All breathing training was performed by a physical therapist with more than 5 years of clinical experience, and all measurements were conducted by a researcher with more than 5 years of clinical experience and a master’s degree or higher under a blinded condition about group assignment and the intervention.

### 2.3. Intervention

Training for voluntary breathing, that is TB and AB, was undertaken using real-time feedback from electromyogram (EMG) signals. During TB training, participants were instructed to: “Breathe to move the upper graph (external intercostal muscle activity) and keep the lower graph (transverse abdominal muscle activity) as flat as possible.”. Similarly, during AB training, participants were instructed to: “Breathe to move the lower graph (transverse abdominal muscle activity) and keep the upper graph (external intercostal muscle activity) as flat as possible.” Respiratory muscles consistent with the breathing type were to be maintained at higher activity than during NB. The rest of the muscles were to be maintained at a flat line with an activity that did not exceed the mean ±2 standard deviations (SDs) of NB activity. After voluntary breathing training, participants were asked to undertake breathing training for 1 week. After 1 week, voluntary breathing activity was assessed and balance ability was measured.

### 2.4. Assessment

#### 2.4.1. Balance Ability

Balance ability was measured with AccuSway (Advanced Mechanical Technology, Inc., Watertown, MA, USA), which showed high reliability (ICC for inter-rater and test-retest reliability = 0.70–0.89) [18]. All outcome variables were processed with Balance Clinic (AMTI, Watertown, MA, USA) and MATLAB (MathWorks, Natick, MA, USA, version R2020b) based on the displacement of the center of pressure. The sampling frequency was set to 200 Hz, and a ‘fourth-order Butterworth’ low-pass filter with a cut-off frequency of 10 Hz was used. All subjects were assessed on their balance ability in the following environments [11]: (1) staring at a X-shaped target located at eye level at a distance of 1.5 m, (2) tips of the toes 30° apart with a distance of 9 cm between the heels; and (3) both hands crossed over the shoulder (Figure 2). For measurement of NB, no specific breathing-related instructions other than the measurement position were provided to avoid focusing on the breathing. During voluntary breathing measurement, participants were instructed to maintain their trained breathing techniques. Balance ability was evaluated in both one-legged and two-legged standing, and each test was repeated three times. Each measurement was performed for 20 s, with a 2-min rest period between the trials. In order to exclude the postural perturbation that occurs immediately after taking up a posture from the data, the middle 10 s of data out of a total of 20 s of measurement were used for analysis. The measured balance variables are shown in Table 1.

#### 2.4.2. Muscle Activity

Real-time feedback via EMG signals was used to evaluate voluntary breathing during breathing training and measurements. EMG signals were collected with AcqKnowledge 5.0 (BIOPAC systems, Goleta, CA, USA). The band-pass filter was set to 30–500, and the sampling rate was set to 1000 Hz. The EMG signal was full-wave rectified. Electrodes were attached to the external intercostal muscles (between the 6th and 7th ribs), sternocleidomastoid muscle (1/3 of the distance between the mastoid process and the jugular notch), transverse abdominis muscle (3 cm lateral to the center of the navel), and rectus abdominis muscle (2 cm below the anterior superior iliac spine) after hair removal and exfoliation with alcohol [19,20].

### 2.5. Statistical Analysis

For all variables, the average value of the data was repeated three times and expressed as the mean ± SD. Statistical analysis was performed using SPSS statistical software (version 25.0; SPSS Inc., Chicago, IL, USA). The Kolmogorov–Smirnov test was used to assess normal distribution, and non-parametric statistical methods were used for non-normal variables. Balance ability with each breathing technique was compared using the Friedman test followed by the Wilcoxon signed-rank test, and the significance level was corrected using the Bonferroni correction method. The critical value of significance was set at α = 0.05.

## 3. Results

### 3.1. General Characteristics of Participants

The 78 subjects who participated in this study completed all the interventions and evaluations; there were no dropouts. The general characteristics of the participants are presented in Table 2.

### 3.2. Electromyogram

Table 3 shows respiratory muscle activity during double-leg stance and single-leg stance. The external intercostals and sternocleidomastoid were significantly more active in TB while the transverse abdominis and rectus abdominis were significantly more active in AB.

### 3.3. Balance Ability

#### 3.3.1. Balance Ability in Double-Leg Stance

The balance ability results for standing on both feet are presented in Table 4. The sway velocity and anterior-posterior difference were significantly smaller in NB than in TB and AB (*p* < 0.05). The anterior-posterior standard deviation was lowest in NB and highest in AB (*p* < 0.05). The anterior-posterior sample entropy was highest in NB and lowest in AB (*p* < 0.05). There was no significant difference between conditions in sway area, right-left difference, right-left standard deviation, and right-left sample entropy.

#### 3.3.2. Balance Ability in Single-Leg Stance

The results of the balance ability on one foot are shown in Table 5. Sway velocity was significantly lower in NB than in TB or AB (*p* < 0.05). The anterior-posterior difference was significantly smaller in NB than in TB or AB (*p* < 0.05). The right-left standard deviation was significantly smaller in NB than in TB or AB (*p* < 0.05). The anterior-posterior standard deviation was lowest in NB and highest in AB (*p* < 0.05). The entropy of the right-left sample was significantly higher in NB than in TB or AB (*p* < 0.05). The anterior-posterior sample entropy was highest in NB and lowest in AB (*p* < 0.05). There was no significant difference between the sway area and the right-left difference between the conditions.

## 4. Discussion

This study investigated the effect of voluntary breathing on the balance ability of healthy, young adults. It found that voluntary breathing affects balance ability in two-legged and one-legged standing in healthy, young adults and that AB has a greater effect than TB. Some previous studies have suggested that voluntary breathing can affect the balance ability of the elderly and patients, and our study also confirmed the same tendency in young adults. These results suggest that voluntary breathing may affect balance ability during static standing in the elderly and patients as well as young healthy adults, and it should be controlled to properly perform balance measurements.

Manor et al. [21] defined the phenomenon of postural sway induced by breathing as ‘posturo-respiratory synchronization’. Also, he suggested that the elderly or patients with physical disorders had a high degree of synchronization between breathing and posture so that postural sway during static standing would be greatly affected by breathing. Our results showed that sway velocity and anterior-posterior standard deviation increased, and sample entropy decreased during voluntary breathing. These results suggest that healthy, young adults may also be affected by posturo-respiratory synchronization. To maintain postural balance, an ankle strategy is used when slow and low-amplitude perturbations occur, and a hip strategy is used when fast and large amplitude perturbations occur [22]. Hernandez et al. [12] suggested that elderly patients with low back pain who have difficulty in tolerating internal perturbation caused by breath could maintain postural balance by using the hip joint strategy in addition to the ankle strategy. The present study found an increase in postural sway in the forward-backward direction, similar to that in the previous study during one-legged standing in healthy, young adults. These results show that voluntary breathing could affect the equilibrium state of healthy adults and that in challenging situations, an increase in postural sway could occur in healthy young adults, similar to that in the elderly and patients with physical disorders.

When postural perturbation occurs, in addition to using a balance strategy to maintain postural stabilization, anticipatory postural adjustments (APA) could minimize the effects of postural perturbation [23]. Additionally, in the case of large postural perturbations that cannot be offset, balance is maintained through compensatory movements [24,25]. A previous study [12] suggested that postural perturbation by respiratory-induced motion was negligible or absent in healthy young adults, whereas individuals with reduced APA, such as the elderly and patients with low back pain, could be affected. However, our results showed that sway velocity and distance in healthy adults increased during voluntary breathing, and that sway variability and irregularity were also affected. These results can be explained by the differences in the mechanisms of neutral and voluntary breathing. Unlike NB controlled by the brainstem, voluntary breathing involves the activation of the cerebral cortex [26]. The change in the activity of the motor area by voluntary breathing can affect the movement of the trunk and extremities [27]. These effects can interfere with the APA ability to maintain balance, even in healthy young adults.

An increase in respiratory muscle activity during voluntary breathing may affect the increase in sway. Voluntary breathing induces greater activation of respiratory muscles than in NB condition, and our results also showed that the activity of external intercostal muscle and sternocleidomastoid muscle was higher in TB than in NB (Table 3). Furthermore, during AB, the activity of the transverse abdominis and rectus abdominis muscles was higher than that under NB condition. To control trunk volume during voluntary breathing, the thoracic respiratory muscles generate movements of the rib cage in the anterior-posterior, lateral, and up-down directions [19,28], and the AB muscles generate an anterior-posterior movement of the abdominal region [20,29]. These additional movements during voluntary breathing could cause internal perturbations and affect the balance ability. In addition, increased activity of the AB muscles can generate torque in the sagittal plane of the trunk and pelvis. This torque in the sagittal plane can increase sway by generating flexion and extension movements. Unlike previous studies that reported that TB had a greater effect on balance ability than AB, our study showing that AB had a greater effect on balance ability is considered to be due to the difference in these breathing mechanisms [13].

Balance ability has been most commonly used to evaluate the motor function of healthy adults, the elderly, athletes, and patients with the disorder in the clinic and various sports environments. In order to perform this evaluation more accurately, a previous study has recommended excluding as much as possible the influence of external factors that may affect postural balance and physical function, and to proceed with the evaluation by adjusting the subject’s foot position, arm posture, gaze, and measurement surface [11]. While previous studies have suggested that postural balance ability occurs due to voluntary breathing only in the population with reduced physical function [12,13], this study demonstrated that balance ability could be influenced by voluntary breathing even in healthy young adults. Our findings suggest that the command to control breathing should also be considered for control body conditions for postural balance assessment, even in the elderly, patients with the disorder, and healthy adults.

This study investigated changes in the balance ability of healthy, young adults during different breathing methods. However, our study has several limitations. First, the sample size was small; therefore, it is difficult to generalize the results. Second, we did not measure the subject’s foot size and width. Therefore, it is difficult to confirm the influence of individual differences according to foot size on postural balance changes following voluntary breathing. Third, when measuring muscle activity, the activity of other muscles was not known because the respiratory muscles were mainly measured. Fourth, it was not possible to control the respiration volume equally during the measurements. For this reason, it was not possible to confirm the difference according to the volume of respiration. Finally, we could not measure changes in posture, such as rib cage volume and changes in three-dimensional movements, during breathing. In order to generalize the study results, it is suggested that additional body measurements, evaluation of extremity muscles, and monitoring of respiratory rate and trunk movement should be performed in subsequent studies.

## 5. Conclusions

The results of this study showed that there is a difference in the balance ability of healthy, young adults according to the type of breathing. In particular, it was found that AB had a greater effect on balance ability than other breathing methods. Therefore, the breathing method should be controlled for balance evaluations.

## Figures and Tables

**Figure 1 ijerph-19-01205-f001:**
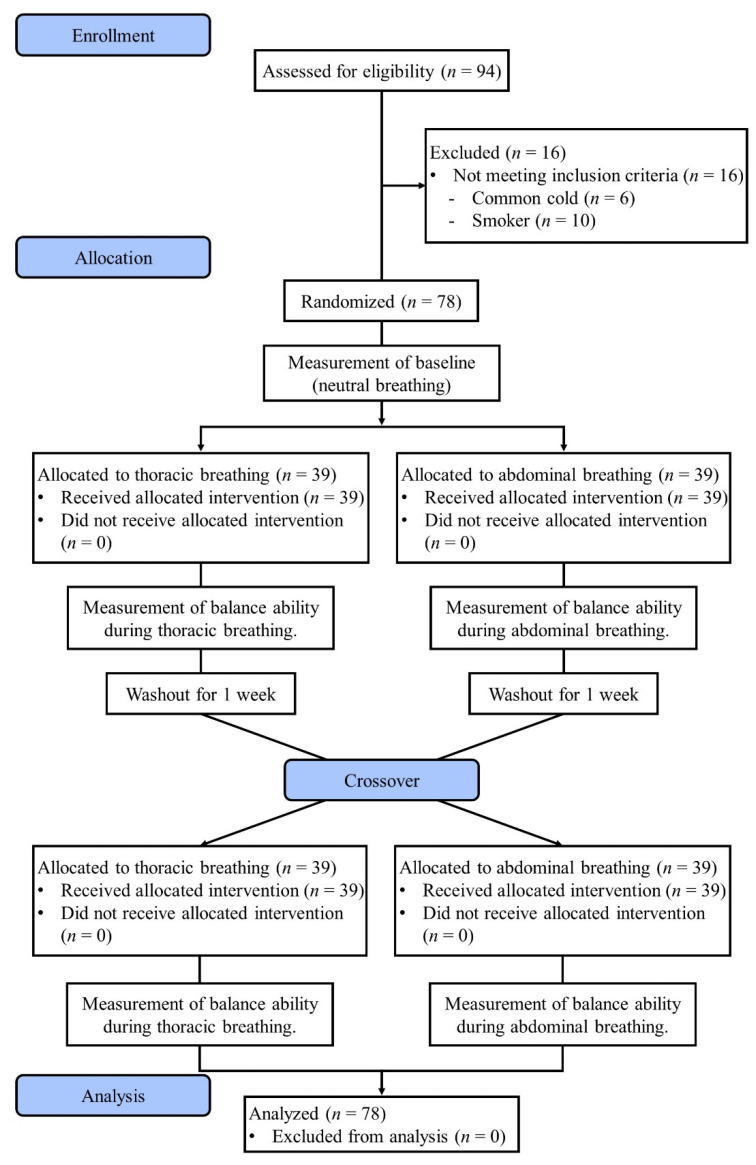
A CONSORT flow diagram with crossover design.

**Figure 2 ijerph-19-01205-f002:**
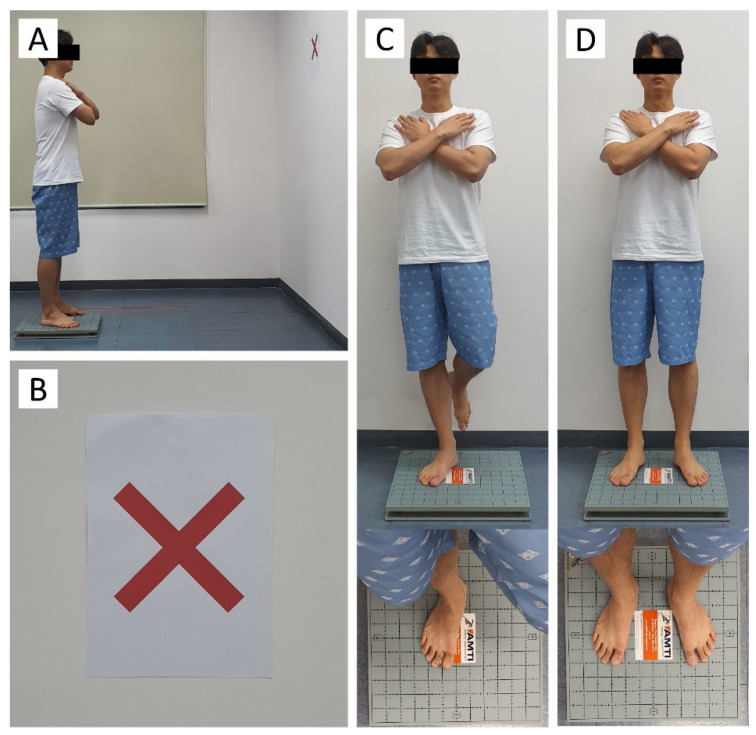
Static balance measurement. (**A**), Balance measuring environment. (**B**), X-shaped target. (**C**), One-legged standing posture. (**D**), Two-legged standing posture.

**Table 1 ijerph-19-01205-t001:** Balance ability parameters.

Variables	Definition
Sway Area	The area of COP displacement per unit time.
Sway Velocity	The movement speed of the COP displacement per unit time.
Anterior-Posterior difference	The maximum distance in the anterior-posterior direction of the COP displacement.
Right-Left difference	The maximum distance in the left-right direction of the COP displacement.
Anterior-Posterior standard deviation	The variability in the anterior-posterior direction of the COP displacement.
Right-Left standard deviation	The variability in the left-right direction of the COP displacement.
Anterior-Posterior sample entropy	The complexity in the anterior-posterior direction of the COP displacement.
Right-Left sample entropy	The complexity in the left-right direction of the COP displacement.

**Table 2 ijerph-19-01205-t002:** General characteristics of participants.

Variables	Participants (*n* = 78)
Sex (female/male) ∗	38/40
Age (years) ^†^	22.54 ± 2.69
Height (cm) ^†^	169.68 ± 8.43
Weight (kg) ^†^	63.82 ± 12.47
Body mass index (kg/m^2^) ^†^	22.01 ± 2.89
Leg length (cm) ^†^	85.99 ± 4.73

∗ Values are expressed as number (N); ^†^ Values are expressed as mean ± SD.

**Table 3 ijerph-19-01205-t003:** Muscle activity according to the type of breathing.

Electromyography	BL	NB M ± SD	TB M ± SD	AB M ± SD	*p*-Value
Double-leg stance (uV)					
External intercostals	5.90	3.70 ± 1.38	6.91 ± 2.81 ***	5.04 ± 0.46 **^†^	<0.001
Sternocleidomastoid	4.81	3.55 ± 1.33	7.01 ± 1.73 ***	3.13 ± 1.59 ^†††^	<0.001
Transverse abdominis	16.06	10.34 ± 2.80	15.71 ± 4.14 **	21.66 ± 1.22 ***^†††^	<0.001
Rectus abdominis	3.31	2.27 ± 0.49	2.32 ± 0.44	5.48 ± 1.30 ***^†††^	<0.001
Single-leg stance (uV)					
External intercostals	7.55	4.86 ± 1.62	8.24 ± 1.58 ***	5.39 ± 1.33 *^†††^	<0.001
Sternocleidomastoid	5.03	4.49 ± 0.23	5.30 ± 0.32 ***	4.82 ± 0.43	<0.001
Transverse abdominis	23.88	17.17 ± 4.55	20.09 ± 6.28 *	26.40 ± 5.79 ***^††^	<0.001
Rectus abdominis	5.03	4.41 ± 2.56	4.12 ± 1.94	9.82 ± 1.98 ***^†††^	<0.001

Note: M, mean; SD, standard deviation; BL, baseline; NB, neutral breathing; TB, thoracic breathing; AB, abdominal breathing; * *p* < 0.05, ** *p* < 0.01, *** *p* < 0.001, vs. NB; ^†^
*p* < 0.05, ^††^
*p* < 0.01, ^†††^
*p* < 0.001, vs. TB.

**Table 4 ijerph-19-01205-t004:** Balance ability in double-leg stance.

Variable	NB		TB		AB		Total-*p*	Post-Hoc Test		
	Mean ± SD	Median (IQR)	Mean ± SD	Median (IQR)	Mean ± SD	Median (IQR)		NB vs. TB	NB vs. AB	TB vs. AB
SA (cm^2^)	1.19 ± 0.81	1.01 (0.88)	1.24 ± 0.80	1.13 (0.78)	1.36 ± 1.01	1.12 (0.96)	0.174	1.000	0.072	0.327
SV (cm/s)	1.51 ± 0.66	1.36 (1.19)	1.59 ± 0.65	1.48 (1.06)	1.63 ± 0.68	1.57 (1.08)	<0.001	0.010	<0.001	0.061
RL-Diff (cm)	0.96 ± 0.58	0.89 (0.73)	0.92 ± 0.56	0.81 (0.58)	0.95 ± 0.63	0.80 (0.58)	0.905	1.000	1.000	1.000
AP-Diff (cm)	2.01 ± 1.29	1.57 (1.73)	2.28 ± 1.35	1.95 (1.63)	2.34 ± 1.38	2.24 (1.62)	<0.001	<0.001	<0.001	0.869
RL-SD	0.180 ± 0.093	0.159 (0.125)	0.181 ± 0.089	0.162 (0.084)	0.180 ± 0.096	0.153 (0.112)	0.752	1.000	1.000	1.000
AP-SD	0.221 ± 0.115	0.218 (0.138)	0.320 ± 0.182	0.306 (0.224)	0.357 ± 0.187	0.337 (0.234)	<0.001	<0.001	<0.001	0.008
RL-SE	0.088 ± 0.037	0.082 (0.038)	0.091 ± 0.035	0.087 (0.042)	0.086 ± 0.028	0.082 (0.037)	0.746	0.825	1.000	0.991
AP-SE	0.079 ± 0.038	0.073 (0.049)	0.063 ± 0.033	0.055 (0.034)	0.057 ± 0.028	0.050 (0.035)	<0.001	<0.001	<0.001	0.033

NB, neutral breathing; TB, thoracic breathing; AB, abdominal breathing; SD, standard deviation; IQR, interquartile range; SA, sway area; SV, sway velocity; RL, right-left; AP, anterior-posterior; Diff, difference; SE, sample entropy.

**Table 5 ijerph-19-01205-t005:** Balance ability in single-leg stance.

Variable	NB		TB		AB		Total-*p*	Post-Hoc Test		
	Mean ± SD	Median (IQR)	Mean ± SD	Median (IQR)	Mean ± SD	Median (IQR)		NB vs. TB	NB vs. AB	TB vs. AB
SA (cm^2^)	9.52 ± 3.61	8.94 (4.80)	9.66 ± 3.55	9.02 (3.88)	9.26 ± 4.23	8.25 (3.02)	0.304	1.000	1.000	0.352
SV (cm/s)	4.72 ± 0.86	4.62 (1.04)	5.34 ± 1.00	5.07 (1.16)	5.28 ± 0.94	4.96 (1.33)	<0.001	<0.001	0.004	0.534
RL-Diff (cm)	3.33 ± 0.80	3.25 (1.10)	3.21 ± 0.82	3.00 (0.93)	3.07 ± 0.81	2.78 (0.97)	0.477	1.000	0.249	0.273
AP-Diff (cm)	9.59 ± 2.62	10.52 (1.17)	10.23 ± 2.65	11.25 (1.31)	10.08 ± 2.62	11.06 (1.00)	<0.001	<0.001	<0.001	0.096
RL-SD	0.566 ± 0.136	0.565 (0.168)	0.624 ± 0.169	0.571 (0.170)	0.627 ± 0.162	0.590 (0.175)	0.001	0.035	0.002	1.000
AP-SD	0.605 ± 0.173	0.550 (0.208)	0.643 ± 0.164	0.636 (0.217)	0.692 ± 0.209	0.616 (0.183)	<0.001	<0.001	< 0.001	0.011
RL-SE	0.093 ± 0.039	0.086 (0.042)	0.075 ± 0.033	0.072 (0.046)	0.075 ± 0.026	0.073 (0.031)	<0.001	<0.001	<0.001	1.000
AP-SE	0.108 ± 0.064	0.090 (0.038)	0.080 ± 0.031	0.074 (0.032)	0.071 ± 0.026	0.073 (0.035)	<0.001	<0.001	<0.001	0.002

NB, neutral breathing; TB, thoracic breathing; AB, abdominal breathing; SD, standard deviation; IQR, interquartile range; SA, sway area; SV, sway velocity; RL, right-left; AP, anterior-posterior; Diff, difference; SE, sample entropy.

## Data Availability

The data presented in this study are available on request from the corresponding author.

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
