# Peer review of "Impact of Types of Breathing on Static Balance Ability in Healthy Adults"

_ijerph, 2022, doi:10.3390/ijerph19031205_

Round 1

Reviewer 1 Report

Abstract: 

1. In the abstract more descriptions regarding the results are warranted.

Introduction: 

1. Need to mention about the magnitude of balance in normal and diseased

2. Background should be more explained with need for the study

Methods:

1. Participants mean age should be mentioned in the participants section

2. Table 1; foot note mean ± SD should be mentioned , the authors have mentioned only +.

3. As the authors have followed the CONSORT guidelines, they should cover all the points in the checklist. However, there was no mentioning of blinding.

4. Justification needed; as its not a randomised controlled trial then what was the need for registering in clinical trial registry.

Results:

1. Though the authors have used Friedmans test and Wilcoxon signed rank test, the results are represented as mean ± SD rather than median and range.

Discussion:

1. More studies against the current study could be discussed depending on the availability of literature

2. Later sections of discussion should cover the clinical implications of this study, how relevant and how much it can be utilised clinically

Author Response

I would like to thank you for providing the opportunity to revise and resubmit the attached manuscript entitled “Impact of Types of Breathing on Static Balance Ability in Healthy” for publication in IJERPH.

We deeply appreciate the editorial comments and reviewers’ helpful comments on our manuscript which we ignored. We agreed with the points addressed by the Reviewers. We provide our responses to the Reviewers’ comments. Please review the attached files.

Reviewer 2 Report

I congratulate the authors for their work and for conducting the experiment. However, the manuscript needs to be improved. I suggest a special effort to improve the rationale and discussion, especially the possible implications of the findings. Some detailed comments are below.

Abstract

Please rewrite the first sentence, since “…or were patients” seems incomplete. The same goes for the end of the introduction.

Please clarify this sentence “In particular, abdominal breathing was more impacted than thoracic breathing (p < 0.05).”. What do you mean by "impacted"?

Introduction

Please, detail the sentence “Hernandez et al. [11] suggested that voluntary breathing affects postural control. Alain et al. [12] reported that thoracic breathing (TB) had a greater effect on postural disturbance than abdominal breathing (AB).”. How postural control is affected? Which disturbances are you refereeing to?

The introduction is very poor. The authors need to provide a better rationale for the study. Why it would be important to study the association between breathing and balance in healthy young people? Please also provide references.

Material and methods

2.4.2. Muscle activity

Please provide more details about EMG measures, such as electrode position.

Discussion

Please review the sentence “It proved that voluntary”. The word “proved” is too strong.

The authors concentrated their efforts in exploring the possible mechanisms that might explain the results. However, it would be important to show the possible implications of the results. Why it would be important to know this information? How it can be used in a real-world situation?

Author Response

(The authors gave the same response as above.)

Reviewer 3 Report

The purpose of this study investigated the differences in balance ability according to the type of breathing in healthy young adults. This study included 78 healthy, young adults. All subjects were assessed for balance ability in neutral breathing, thoracic breathing, and abdominal breathing through a crossover design. Balance ability was assessed during static standing using a force plate. The results of this study showed that there is a difference in the balance ability of healthy, young adults according to the type of breathing. In particular, abdominal breathing had a greater effect on balance ability than other breathing methods. Overall, the authors make evidence-based claims with relatively appropriate study designs. However, the following concerns should be addressed.

  1. Special issue is related to sedentary lifestyle. What is the level of sedentary lifestyle of the subjects? For example, the subjects' average sedentary time, average exercise level, etc.
  2. What is the subject's foot size?
  3. For a better understanding, please present figures on how to do static balance measuring posture for double- and single-leg stances, foot posture, upper extremity posture, and screen location.
  4. In the method, please describe the meaning of increase or decrease in the values ​​of sway area, sway velocity, right-left difference, anterior-posterior difference, and sample entropy.
  5. Has the balance evaluation method of this study been validated in previous studies?
  6. Balance ability is a key factor of this study. Who and how many authors measured balance ability?
  7. ICC may vary from study to study. What is the ICC for inter-rater reliability and test-retest reliability in this study?
  8. Please describe the limitations of this study in more detail. In particular, include the authors' efforts to address these limitations.

Author Response

(The authors gave the same response as above.)

Round 2

Reviewer 1 Report

The authors have answered all the queries and replied to the comments in a satisfactory manner 

Author Response

Dear Reviewer,

I would like to thank you for providing the opportunity to revise and resubmit the attached manuscript entitled “Impact of Types of Breathing on Static Balance Ability in Healthy” for publication in IJERPH.

We deeply appreciate reviewer’ helpful comments on our manuscript which we ignored. We agreed with the points addressed by the Reviewer, and provide our responses to your comments. Please review the attached file.
